# Automating Nearest Neighbor Search Configuration with Constrained Optimization

**Philip Sun, Ruiqi Guo & Sanjiv Kumar**
Google Research
{sunphil,guorq,sanjivk}@google.com

## Abstract

The approximate nearest neighbor (ANN) search problem is fundamental to efficiently serving many real-world machine learning applications. A number of techniques have been developed for ANN search that are efficient, accurate, and scalable. However, such techniques typically have a number of parameters that affect the speed-recall tradeoff, and exhibit poor performance when such parameters aren't properly set. Tuning these parameters has traditionally been a manual process, demanding in-depth knowledge of the underlying search algorithm. This is becoming an increasingly unrealistic demand as ANN search grows in popularity. To tackle this obstacle to ANN adoption, this work proposes a constrained optimization-based approach to tuning quantization-based ANN algorithms. Our technique takes just a desired search cost or recall as input, and then generates tunings that, empirically, are very close to the speed-recall Pareto frontier and give leading performance on standard benchmarks.

## 1 Introduction

Efficient nearest neighbor search is an integral part of approaches to numerous tasks in machine learning and information retrieval; it has been leveraged to effectively solve a number of challenges in recommender systems (Benzi et al., 2016; Cremonesi et al., 2010), coding theory (May & Ozerov, 2015), multimodal search (Gfeller et al., 2017; Miech et al., 2021), and language modeling (Guu et al., 2020; Khandelwal et al., 2020; Kitaev et al., 2020). Vector search over the dense, high-dimensional embedding vectors generated from deep learning models has become especially important following the rapid rise in capabilities and performance of such models. Nearest neighbor search is also increasingly being used for assisting training tasks in ML (Lindgren et al., 2021; Yen et al., 2018).

Formally, the nearest neighbor search problem is as follows: we are given an $n$-item dataset $\mathcal{X} \in \mathbb{R}^{n \times d}$ composed of $d$-dimensional vectors, and a function for computing the distance between two vectors $D : \mathbb{R}^d \times \mathbb{R}^d \mapsto \mathbb{R}$. For a query vector $q \in \mathbb{R}^d$, our goal is to find the indices of the $k$-nearest neighbors in the dataset to $q$:

$$k\text{-}\underset{i \in \{1,\dots,n\}}{\arg\min} D(q, \mathcal{X}_i)$$

Common choices of $D$ include $D(q, x) = -\langle q, x \rangle$ for maximum inner product search (MIPS) and $D(q, x) = \|q - x\|_2^2$ for Euclidean distance search. A linear-time scan over $\mathcal{X}$ solves the nearest neighbor search problem but doesn't scale to the large dataset sizes often found in modern-day applications, hence necessitating the development of approximate nearest neighbor (ANN) algorithms.

A number of approaches to the ANN problem have been successful in trading off a small search accuracy loss, measured in result recall, for a correspondingly large increase in search speed (Aumüller et al., 2020). However, these approaches rely on tuning a number of hyperparameters that adjust the tradeoff between speed and recall, and poor hyperparameter choices may result in performance far below what could be achievable with ideal hyperparameter tuning. This tuning problem becomes especially difficult at the billions-scale, where the larger dataset size typically leads to a greater number of hyperparameters to tune. Existing approaches to tuning an ANN index, enumerated in Table 1, all suffer from some deficiency, such as using an excessive amount of computation

Table 1: Our technique is the first to use minimal computational cost and human involvement to configure an ANN index to perform very close to its speed-recall Pareto frontier.

| Method | Computational Cost of Tuning | Human Involvement | Hyperparameter Quality |
|---|---|---|---|
| Grid search | High | **Low** | **High** |
| Manual tuning | **Low** | High | Medium |
| Black-box optimizer | Medium | **Low** | Medium |
| Ours | **Low** | **Low** | **High** |

during the tuning process, necessitating extensive human-in-the-loop expertise, or giving suboptimal hyperparameters.

Mitigating these issues is becoming increasingly important with the growth in dataset sizes and in the popularity of the ANN-based retrieval paradigm. This paper describes how highly performant ANN indices may be created and tuned with minimal configuration complexity to the end user. Our contributions are:

- Deriving theoretically-grounded models for recall and search cost for quantization-based ANN algorithms, and presenting an efficient Lagrange multipliers-based technique for optimizing either of these metrics with respect to the other.

- Showing that on millions-scale datasets, the tunings from our technique give almost identical performance to optimal hyperparameter settings found through exhaustive grid search.

- Achieving superior performance on track 1 of the billions-scale `big-ann-benchmarks` datasets using tunings from our technique over tunings generated by a black-box optimizer on the same ANN index, and over all existing benchmark submissions.

Our constrained optimization approach is very general and we anticipate it can be extended to distance measures, quantization algorithms, and search paradigms beyond those explored in this paper.

## 2  RELATED WORK

### 2.1  ANN ALGORITHMS

Literature surrounding the ANN problem is extensive, and many solutions have been proposed, drawing inspiration from a number of different fields. Below we give a brief outline of three families of approaches that have found empirical success and continued research interest, with an emphasis on the hyperparameters necessitated by each approach. Other approaches to ANN include sampling-based algorithms (Liu et al., 2019) and a variety of geometric data structures (Bozkaya & Ozsoyoglu, 1997; Ram & Gray, 2012); we refer readers to Bhatia & Vandana (2010); RezaAbbasifard et al. (2014); Wang et al. (2014; 2021) for more comprehensive surveys.

**Hashing approaches**   Techniques under this family utilize locality sensitive hash (LSH) functions, which are functions that hash vectors with the property that more similar vectors are more likely to collide in hash space (Andoni & Razenshteyn, 2015; Datar et al., 2004; Shrivastava & Li, 2014). By hashing the query and looking up the resulting hash buckets, we may expect to find vectors close to the query. Hashing algorithms are generally parameterized by the number and size of their hash tables. The random memory access patterns of LSH often lead to difficulties with efficient implementation, and the theory that prescribes hyperparameters for LSH-based search generally cannot consider dataset-specific idiosyncrasies that allow for faster search than otherwise guaranteed for worst-case inputs; see Appendix A.1 for further investigation.

**Graph approaches**   These algorithms compute a (potentially approximate) nearest neighbor graph on $\mathcal{X}$, where each element of $\mathcal{X}$ becomes a graph vertex and has directed edges towards its nearest neighbors. The nearest neighbors to $q$ are computed by starting at some vertex and traversing edges

to vertices closer to $q$. These algorithms are parameterized by graph construction details, such as the number of edges; any post-construction adjustments, such as improving vertex degree distribution and graph diameter (Fu et al., 2019; Iwasaki & Miyazaki, 2018; Malkov & Yashunin, 2020); and query-time parameters for beam search and selecting the initial set of nodes for traversal.

**Quantization approaches**   These algorithms create a compressed form of the dataset $\tilde{\mathcal{X}}$; at query time, they return the points whose quantized representations are closest to the query. Speedups are generally proportional to the reduction in dataset size. These reductions can be several orders of magnitude, although coarser quantizations lead to greater recall loss. Quantization techniques include VQ, where each datapoint is assigned to the closest element in a codebook $C$, and a number of multi-codebook quantization techniques, where the datapoint is approximated as some aggregate (concatenation, addition, or otherwise) of the codebook element it was assigned to per-codebook (Babenko & Lempitsky, 2015; Guo et al., 2020; Jégou et al., 2011). Quantization-based approaches generally must be tuned on codebook size and the number of codebooks.

**Growth in parameterization complexity with respect to dataset size**   While the above approaches may only each introduce a few hyperparameters, many of the best-performing ANN algorithms layer multiple approaches, leading to a higher-dimensional hyperparameter space much more difficult to tune. For example, Chen et al. (2021) uses VQ, but also a graph-based approach to search over the VQ codebook. Other algorithms (Guo et al., 2020; Johnson et al., 2021), including what we discuss in this work, use VQ to perform a first-pass pruning over the dataset and then a multi-codebook quantization to compute more accurate distance estimates.

## 2.2 ANN Hyperparameter Tuning

Tuning in low-dimensional hyperparameter spaces may be effectively handled with grid search; however, quantization-based ANN algorithms with few hyperparameters scale poorly with dataset size, as shown in Appendix A.8. In higher-dimensional ANN hyperparameter spaces, where grid search is computationally intractable, there are two predominant approaches to the tuning problem, each with their drawbacks. The first is using heuristics to reduce the search space into one tractable with grid search, as in Criteo (2021) or Ren et al. (2020). These heuristics may perform well when set and adjusted by someone with expertise in the underlying ANN search algorithm implementation, but such supervision is often impractical or expensive; otherwise, these heuristics may lead to suboptimal hyperparameter choices.

The second approach is to use black-box optimization techniques such as Bergstra et al. (2011) or Golovin et al. (2017) to select hyperparameters in the full-dimensionality tuning space. These algorithms, however, lack inductive biases from knowing the underlying ANN search problem and therefore may require a high number of samples before finding hyperparameter tunings that are near optimal. Measurement noise from variability in machine performance further compounds the challenges these black-box optimizers face.

## 3 Preliminaries

### 3.1 Large-Scale Online Approximate Nearest Neighbors

In this work we focus on the problem of tuning ANN algorithms for online search of large-scale datasets. In this scenario, the search algorithm must respond to an infinite stream of latency-sensitive queries arriving at roughly constant frequency. This setting is common in recommender and semantic systems where ANN speed directly contributes to the end-user experience. We are given a sample set of queries, representative of the overall query distribution, with which we can tune our data structure.

Large dataset size makes the linear scaling of exact brute-force search impractical, so approximate search algorithms must be used instead. These algorithms may be evaluated along two axes:

1. **Accuracy**: quantified by recall@$k$, where $k$ is the desired number of neighbors. Sometimes the $c$-approximation ratio, the ratio of the approximate and the true nearest-neighbor distance, is used instead; we correlate between this metric and recall in Appendix A.1.
2. **Search cost**: typically quantified by the queries per second (QPS) a given server can handle.

An effective ANN solution maximizes accuracy while minimizing search cost.

## 3.2 Vector Quantization (VQ)

The ANN algorithm we tune uses a hierarchical quantization index composed of vector quantization (VQ) and product quantization (PQ) layers. We first give a brief review of VQ and PQ before describing how they are composed to produce a performant ANN search index.

Vector-quantizing an input set of vectors $\mathcal{X} \in \mathbb{R}^{n \times d}$, which we denote $VQ(X)$, produces a codebook $C \in \mathbb{R}^{c \times d}$ and codewords $w \in \{1, 2, \ldots, c\}^n$. Each element of $X$ is quantized to the closest codebook element in $C$, and the quantization assignments are stored in $w$. The quantized form of the $i$th element of $\mathcal{X}$ can therefore be computed as $\tilde{\mathcal{X}}_i = C_{w_i}$.

VQ may be used for ANN by computing the closest codebook elements to the query

$$\mathcal{S} := k\text{-}\arg\min_{i \in \{1, 2, \ldots, c\}} D(q, C_i)$$

and returning indices of datapoints belonging to those codebook elements, $\{j | w_j \in \mathcal{S}\}$. This candidate set may also be further refined by higher-bitrate distance calculations to produce a final result set. In this manner, VQ can be interpreted as a pruning tree whose root stores $C$ and has $c$ children; the $i$th child contains the points $\{\mathcal{X}_j | w_j = i\}$; equivalently, this tree is an inverted index (or inverted file index, IVF) which maps each centroid to the datapoints belonging to the centroid.

## 3.3 Product Quantization (PQ)

Product quantization divides the full $d$-dimensional vector space into $K$ subspaces and quantizes each space separately. If we assume the subspaces are all equal in dimensionality, each covering $l = \lceil d/K \rceil$ dimensions, then PQ gives $K$ codebooks $C^{(1)}, \ldots, C^{(K)}$ and $K$ codeword vectors $w^{(1)}, \ldots, w^{(K)}$, with $C^{(k)} \in \mathbb{R}^{c_k \times l}$ and $w^{(k)} \in \{1, \ldots, c_i\}^n$ where $c_k$ is the number of centroids in subspace $k$. The $i$th element can be recovered as the concatenation of $\{C^{(k)}_{w_i^{(k)}} | k \in \{1, \ldots, K\}\}$.

For ANN search, VQ is generally performed with a large codebook whose size scales with $n$ and whose size is significant relative to the size of the codewords. In contrast, PQ is generally performed with a constant, small $c_k$ that allows for fast in-register SIMD lookups for each codebook element, and its storage cost is dominated by the codeword size.

## 3.4 ANN Search with Multi-Level Quantization

VQ and PQ both produce fixed-bitrate encodings of the original dataset. However, in a generalization of Guo et al. (2020), we would like to allocate more bitrate to the vectors closer to the query, which we may achieve by using multiple quantization levels and using the lower-bitrate levels to select which portions of the higher-bitrate levels to evaluate.

To generate these multiple levels, we start with the original dataset $\mathcal{X}$ and vector-quantize it, resulting in a smaller $d$-dimensional dataset of codewords $C$. We may recursively apply VQ to $C$ for arbitrarily many levels. $\mathcal{X}$ and all $C$ are product-quantized as well. As a concrete example, Figure 6 describes the five-quantization-level setups used in Section 5.2.

This procedure results in a set of quantizations $\tilde{\mathcal{X}}_1, \ldots, \tilde{\mathcal{X}}_m$ of progressively higher bitrate. Algorithm 1 performs ANN using these quantizations and a length-$m$ vector of search hyperparameters $t$, which controls how quickly the candidate set of neighbors is narrowed down while iterating through the quantization levels. Our goal is to find $t$ that give excellent tradeoffs between search speed and recall.

## 4 Method

The following sections derive proxy metrics for ANN recall and search latency as a function of the tuning $t$, and then describe a Lagrange multipliers-based approach to efficiently computing $t$ to optimize for a given speed-recall tradeoff.

---

**Algorithm 1** Quantization-Based ANN

---

1: **procedure** QUANTIZEDSEARCH($\tilde{\mathcal{X}}, t, q$)          ▷ Computes the $t_m$ nearest neighbors to $q$
2:     $\mathcal{S}_0 \leftarrow \{1, \ldots, n\}$
3:     **for** $i \leftarrow 1$ to $m$ **do**              ▷ Iterate over quantizations in ascending bitrate order
4:         $\mathcal{S}_i \leftarrow t_i\text{-}\arg\min_{j \in \mathcal{S}_{i-1}} D(q, \tilde{X}_j^{(i)})$          ▷ Narrow candidate set to $t_i$ elements, using $\tilde{\mathcal{X}}^{(i)}$
5:     **end for**
6:     **return** $\mathcal{S}_m$
7: **end procedure**

---

## 4.1 Proxy Loss for ANN Recall

For a given query set $\mathcal{Q}$ and hyperparameter tuning $t$, the recall may be computed by simply performing approximate search over $\mathcal{Q}$ and computing the recall empirically. However, such an approach has no underlying mathematical structure that permits efficient optimization over $t$. Below we approximate this empirical recall in a manner amenable to our constrained optimization approach.

First fix the dataset $\mathcal{X}$ and all quantizations $\tilde{\mathcal{X}}^{(i)}$. Define functions $\mathcal{S}_0(q, t), \ldots, \mathcal{S}_m(q, t)$ to denote the various $\mathcal{S}$ computed by Algorithm 1 for query $q$ and tuning $t$, and let $\mathcal{G}(q)$ be the set of ground-truth nearest neighbors for $q$. Note our recall equals $\dfrac{|\mathcal{S}_m(q, t) \cap \mathcal{G}(q)|}{|\mathcal{G}(q)|}$ for a given query $q$ and tuning $t$. We can decompose this into a telescoping product and multiply it among all queries in $\mathcal{Q}$ to derive the following expression for geometric-mean recall:

$$\text{GeometricMeanRecall}(\mathcal{Q}, t) = \prod_{q \in \mathcal{Q}} \prod_{i=1}^{m} \left( \frac{|\mathcal{S}_i(q, t) \cap \mathcal{G}(q)|}{|\mathcal{S}_{i-1}(q, t) \cap \mathcal{G}(q)|} \right)^{1/|\mathcal{Q}|}, \tag{1}$$

where the telescoping decomposition takes advantage of the fact that $|\mathcal{S}_0(q, t) \cap \mathcal{G}(q)| = |\mathcal{G}(q)|$ due to $\mathcal{S}_0$ containing all datapoint indices. We choose the geometric mean, despite the arithmetic mean's more frequent use in aggregating recall over a query set, because the geometric mean allows for the decomposition in log-space that we perform below. Note that the arithmetic mean is bounded from below by the geometric mean.

Maximizing Equation 1 is equivalent to minimizing its negative logarithm:

$$\begin{aligned}
\mathcal{L}(\mathcal{Q}, t) &= -\frac{1}{|\mathcal{Q}|} \sum_{q \in \mathcal{Q}} \sum_{i=1}^{m} \log \frac{|\mathcal{S}_i(q, t) \cap \mathcal{G}(q)|}{|\mathcal{S}_{i-1}(q, t) \cap \mathcal{G}(q)|} \\
&= \sum_{i=1}^{m} \mathbb{E}_{q \in \mathcal{Q}} \left[ -\log \frac{|\mathcal{S}_i(q, t) \cap \mathcal{G}(q)|}{|\mathcal{S}_{i-1}(q, t) \cap \mathcal{G}(q)|} \right]
\end{aligned} \tag{2}$$

Now we focus on the inner quantity inside the logarithm and how to compute it efficiently. The chief problem is that $\mathcal{S}_i(q, t)$ has an implicit dependency on $\mathcal{S}_{i-1}(q, t)$ because $\mathcal{S}_{i-1}$ is the candidate set from which we compute quantized distances using $\tilde{\mathcal{X}}^{(i)}$ in Algorithm 1. This results in $\mathcal{S}_i(q, t)$ depending on all $t_1, \ldots, t_i$ and not just $t_i$ itself, making it difficult to efficiently evaluate. To resolve this, define the *single-layer candidate set*

$$\mathcal{S}_i'(q, t_i) = t_i\text{-}\arg\min_{j \in \{1, \ldots, n\}} D(q, \tilde{\mathcal{X}}_j^{(i)}) \tag{3}$$

which computes the closest $t_i$ neighbors to $q$ according to only $\tilde{\mathcal{X}}^{(i)}$, irrespective of other quantizations or their tuning settings. We leverage this definition by rewriting our cardinality ratio as

$$\frac{|\mathcal{S}_i(q, t) \cap \mathcal{G}(q)|}{|\mathcal{S}_{i-1}(q, t) \cap \mathcal{G}(q)|} = \frac{\sum_{g \in \mathcal{G}(q)} \mathbb{1}_{g \in \mathcal{S}_i(q, t)}}{\sum_{g \in \mathcal{G}(q)} \mathbb{1}_{g \in \mathcal{S}_{i-1}(q, t)}} \tag{4}$$

and making the approximation $\mathbb{1}_{g\in\mathcal{S}_i(q,t)} \approx \mathbb{1}_{g\in\mathcal{S}_{i-1}(q,t)}\mathbb{1}_{g\in\mathcal{S}'_i(q,t_i)}$. This is roughly equivalent to assuming most near-neighbors to $q$ are included in $\mathcal{S}_{i-1}(q,t)$; see Appendix A.2 for further discussion. If we furthermore assume zero covariance[1] between $\mathbb{1}_{g\in\mathcal{S}_{i-1}(q,t)}$ and $\mathbb{1}_{g\in\mathcal{S}'_i(q,t_i)}$, then we can transform the sum of products into a product of sums:

$$\sum_{g\in\mathcal{G}(q)} \mathbb{1}_{g\in\mathcal{S}_{i-1}(q,t)}\mathbb{1}_{g\in\mathcal{S}'_i(q,t_i)} \approx \left(\frac{1}{|\mathcal{G}(q)|}\sum_{g\in\mathcal{G}(q)}\mathbb{1}_{g\in\mathcal{S}_{i-1}(q,t)}\right)\left(\sum_{g\in\mathcal{G}(q)}\mathbb{1}_{g\in\mathcal{S}'_i(q,t_i)}\right).$$

Combining this result from Equations 2 and 4, our final loss function is $\sum_{i=1}^m \mathcal{L}_i(\mathcal{Q},t_i)$, with the *per-quantization loss* $\mathcal{L}_i$ defined as

$$\mathcal{L}_i(\mathcal{Q},t_i) = \mathbb{E}_{q\in\mathcal{Q}}\left[-\log\frac{|\mathcal{S}'_i(q,t_i)\cap\mathcal{G}(q)|}{|\mathcal{G}(q)|}\right]. \tag{5}$$

See Appendix A.4 for how $\mathcal{L}_i$ may be efficiently computed over $\mathcal{Q}$ for all $i\in\{1,\ldots,m\}, t_i\in\{1,\ldots,n\}$, resulting in a matrix $L\in\mathbb{R}^{m\times n}$. This allows us to compute the loss for any tuning $t$ by summing $m$ elements from $L$.

## 4.2 Proxy Metric for ANN Search Cost

Similar to ANN recall, search cost may be directly measured empirically, but below we present a simple yet effective search cost proxy compatible with our Lagrange optimization method.

Let $|\tilde{\mathcal{X}}^{(i)}|$ denote the storage footprint of $\tilde{\mathcal{X}}^{(i)}$. At quantization level $i$, for $i < m$, selecting the top top $t_i$ candidates necessarily implies that a $t_i/n$ proportion of $\tilde{\mathcal{X}}^{(i+1)}$ will need to be accessed in the next level. Meanwhile, $\tilde{\mathcal{X}}^{(1)}$ is always fully searched because it's encountered at the beginning of the search process, where the algorithm has no prior on what points are closest to $q$. From these observations, we can model the cost of quantization-based ANN search with a tuning $t$ as

$$J(t) \triangleq \frac{1}{|\mathcal{X}|}\cdot\left(|\tilde{\mathcal{X}}^{(1)}| + \sum_{i=1}^{m-1}\frac{t_i}{n}\cdot|\tilde{\mathcal{X}}^{(i+1)}|\right). \tag{6}$$

$J$ gives the ratio of memory accesses performed per-query when performing approximate search with tuning $t$ to the number of memory accesses performed by exact brute-force search. This gives a good approximation to real-world search cost because memory bandwidth is the bottleneck for quantization-based ANN in the non-batched case. We emphasize that this cost model is effective for comparing amongst tunings for a quantization-based ANN index, which is sufficient for our purposes, but likely lacks the power to compare performance among completely different ANN approaches, like graph-based solutions. Differences in memory read size, memory request queue depth, amenability to vectorization, and numerous other characteristics have a large impact on overall performance but are not captured in this model. Our model is, however, compatible with query batching, which we discuss further in Appendix A.3.

## 4.3 Convexification of the Loss

We take the convex hull of each per-quantization loss $\mathcal{L}_i$ before passing it into the constrained optimization procedure. This results in a better-behaved optimization result but is also justified from an ANN algorithm perspective. For any quantization level $i$, consider some two choices of $t_i$ that lead to loss and cost contributions of $(l_1, j_1)$ and $(l_2, j_2)$. Any (loss, cost) tuple on the line segment between these two points can be achieved via a randomized algorithm that picks between our two choices of $t_i$ with the appropriate weighting, which implies the entire convex hull is achievable. Empirically, we find that $\mathcal{L}_i$ is extremely close to convex already, so this is more of a theoretical safeguard than a practical concern.

---

[1]Note that $\mathcal{S}_i \subseteq \mathcal{S}_{i-1}$ so this is emphatically false for $\mathbb{1}_{g\in\mathcal{S}_{i-1}(q,t)}$ and $\mathbb{1}_{g\in\mathcal{S}_i(q,t)}$, but with $\mathbb{1}_{g\in\mathcal{S}'_i(q,t_i)}$ we may reasonably assume little correlation with $\mathbb{1}_{g\in\mathcal{S}_{i-1}(q,t)}$.

### 4.4 Constrained Optimization

Formally, our tuning problem of maximizing recall with a search cost limit $J_{\max}$ can be phrased as

$$\underset{t \in [0,n]^m}{\arg\min} \quad \sum_{i=1}^{m} \mathcal{L}_i(t_i)$$
$$\text{s.t.} \quad J(t) \leq J_{\max}$$
$$t_1 \geq \ldots \geq t_m.$$

The objective function is a sum of convex functions and therefore convex itself, while the constraints are linear and strictly feasible, so strong duality holds. We can therefore utilize the Lagrangian

$$\underset{t \in [0,n]^m}{\arg\min} \quad -\lambda J(t) + \sum_{i=1}^{m} \mathcal{L}_i(t_i)$$
$$\text{s.t.} \quad t_1 \geq \ldots \geq t_m.$$

to find exact solutions to the constrained optimization, using $\lambda$ to adjust the recall-cost tradeoff. We show in Appendix A.5 an algorithm that uses $O(nm)$ preprocessing time to solve the minimization for a given value of $\lambda$ in $O(m \log n)$ time.

Furthermore, because the objective function is a sum of $m$ functions, each a convex hull defined by $n$ points, the Pareto frontier itself will be piecewise, composed of at most $nm$ points. It follows then that there are at most $nm$ relevant $\lambda$ that result in different optimization results, namely those obtained by taking the consecutive differences among each $\mathcal{L}_i$ and dividing by $|\tilde{\mathcal{X}}^{(i+1)}|/n|\mathcal{X}|$. By performing binary search among these candidate $\lambda$, we can find the minimum-cost tuning for a given loss target, or the minimum-loss tuning for a given cost constraint, in $O(m \log n \log nm)$ time.

In practice, even for very large datasets, $m < 10$, so this routine runs very quickly. The constrained optimizations used to generate the tunings in Section 5.2 ran in under one second on a Xeon W-2135; computation of $\mathcal{L}$ contributed marginally to the indexing runtime, described further in Appendix A.6.

## 5 Experiments

### 5.1 Million-Scale Benchmarks and Comparison to Grid Search

To see how closely our algorithm's resulting hyperparameter settings approach the true optimum, we compare its output to tunings found through grid search. We compare against the grid-searched parameters used by ScaNN (Guo et al., 2020) in its leading performance on the public `Glove1M` benchmark from Aumüller et al. (2020). As shown in Figure 1 below, our method's tunings are almost exactly on the speed-recall frontier.

While the resulting tunings are of roughly equivalent quality, grid search takes far longer to identify such tunings; it searched 210 configurations in 22 minutes. In comparison, on the same machine, our method took 53 seconds to compute $\mathcal{L}$ and run the constrained optimization to generate tunings.

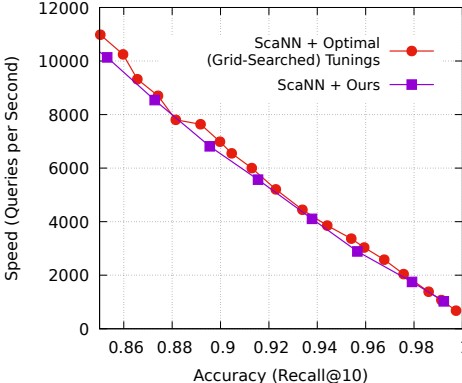

Figure 1: Our technique's tunings compared to grid-searched tunings, applied to ScaNN on `Glove1M`.

### 5.2 Billion-Scale Benchmarks

We now proceed to larger-scale datasets, where the tuning space grows significantly; the following benchmarks use a five-level quantization index (see Appendix A.6.1 for more details), resulting

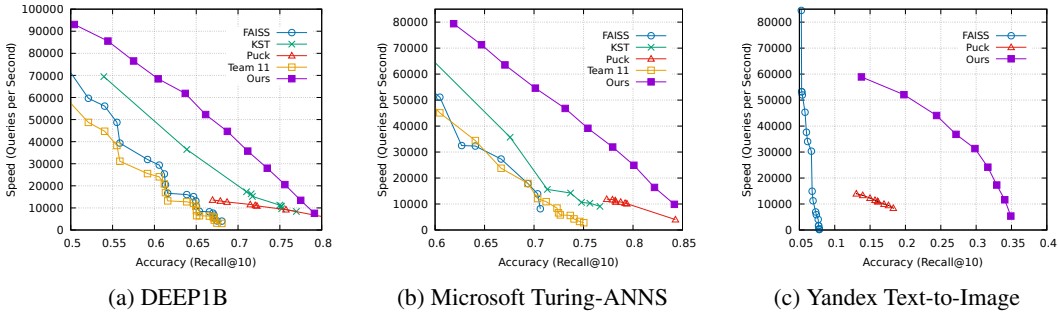

|  |  |  |
|---|---|---|
| (a) DEEP1B | (b) Microsoft Turing-ANNS | (c) Yandex Text-to-Image |

Figure 2: Speed-recall tradeoffs of our tuning algorithm plus ANN search implementation, compared to others from track 1 of the standardized `https://big-ann-benchmarks.com` datasets.

in a four-dimensional hyperparameter space. Even with very optimistic assumptions, this gives hundreds of thousands of tunings to grid search over, which is computationally intractable, so we compare to heuristic, hand-tuned, and black-box optimizer settings instead. Here we use our own implementation of a multi-level quantization ANN index, and benchmark on three datasets from `big-ann-benchmarks.com` (Simhadri et al., 2022), following the experimental setup stipulated by track 1 of the competition; see Appendix A.6 for more details. Our results are shown in Figure 2.

We find that even in this much more complex hyperparameter space, our technique manages to find settings that make excellent speed-recall tradeoffs, resulting in leading performance on these datasets.

### 5.2.1 COMPARISON AGAINST BLACK-BOX OPTIMIZERS

To see if black-box optimizers can effectively tune quantization-based ANN indices, we used Vizier (Golovin et al., 2017) to tune the same DEEP1B index used above. Our Vizier setup involved an in-the-loop ANN index serving online requests, with Vizier generating candidate configurations, measuring their resulting recall and throughput, and then using those measurements to inform further candidate selections. We ran the Vizier study for 6 hours, during which it conducted over 1800 trials; their recall and performance measurements are plotted to the right in Figure 3.

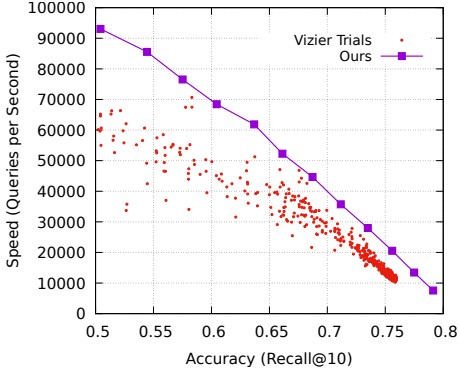

Figure 3: On DEEP1B, our hyperparameter tunings achieve significantly better speed-recall tradeoffs than those found by Vizier.

We can see that Vizier found several effective tunings close to the Pareto frontier of our technique, but then failed to interpolate or extrapolate from those tunings. Our technique not only generated better tunings, but did so using less compute; notably, the computation of statistics $\mathcal{L}_i$ and the constrained optimization procedure were done with low-priority, preemptible, batch compute resources, while in contrast Vizier requires instances of online services in a carefully controlled environment in order to get realistic and low-variance throughput measurements.

### 5.3 RECALL AND COST MODEL ACCURACY

We desire linear, predictable relationships between the modeled and the corresponding real-world values for both recall and search cost. This is important both so that the optimization can produce tunings that are effective in practice, and so that users can easily interpret and apply the results.

In Figure 4, we take the DEEP1B hyperparameter tunings used above in Section 5.2 and plot their respective modeled recalls against empirical recall@10, and modeled search costs against measured reciprocal throughput. Recall was modeled as $\exp(-\sum \mathcal{L}_i)$ while $J$ was simply used for cost. We can conclude from the square of their sample Pearson correlation coefficients ($r^2$) that both relationships between analytical values and their empirical measurements are highly linear.

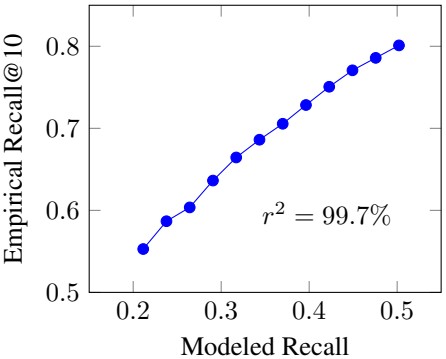 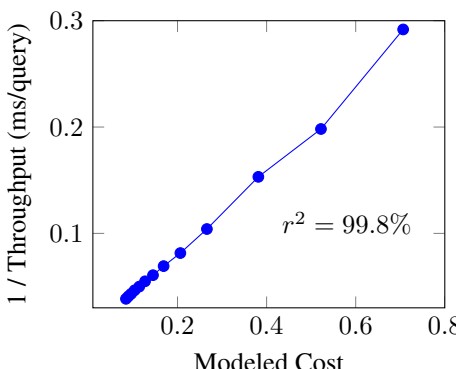

Figure 4: Modeled costs and recalls have a highly linear relationship with their true values.

### 5.4 OUT-OF-SAMPLE QUERY PERFORMANCE

Our hyperparameter tunings were optimized based on statistics calculated on a query sample $\mathcal{Q}$, but in order to fulfill their purpose, these tunings must generalize and provide good performance and recall on the overall query stream. We test generalization capability by randomly splitting the $10^4$ queries in the DEEP1B dataset into two equal-sized halves $\mathcal{Q}_1$ and $\mathcal{Q}_2$. Then we compare the resulting Pareto frontiers of training on $\mathcal{Q}_1$ and testing on $\mathcal{Q}_2$ (out-of-sample), versus training and testing both on $\mathcal{Q}_2$ (in-sample). The resulting Pareto frontiers, shown in Figure 5, are near-indistinguishable and within the range of measurement error from machine performance fluctuations, indicating excellent generalization. Qualitatively, the in-sample and out-of-sample tuning parameters differ only very slightly, suggesting our optimization is robust.

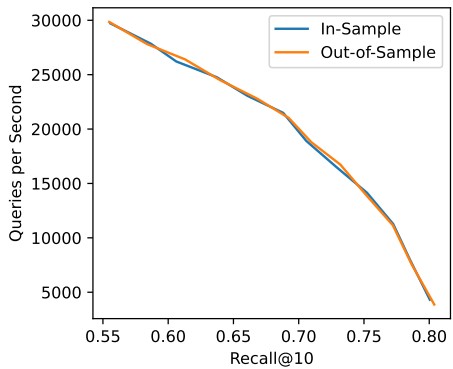

Figure 5: Speed-recall frontiers for tunings derived from in-sample and out-of-sample query sets on the DEEP1B dataset.

### 5.5 ADDITIONAL EXPERIMENTS

In Appendix A.7 we analyze the effects of query sample size and find even a small (1000) query sample is sufficient to provide effective tuning results. Appendix A.8 shows that grid-searching a shallow quantization index (similar to what was done in Section 5.1) fails to perform well on billion-scale datasets.

## 6 CONCLUSION

As adoption of nearest neighbor search increases, so does the importance of providing ANN frameworks capable of providing good performance even to non-experts unfamiliar with the inner details of ANN algorithms. We believe our work makes a significant step towards this goal by providing a theoretically grounded, computationally efficient, and empirically successful method for tuning quantization-based ANN algorithms. However, our tuning model still relies on the user to pick the VQ codebook size, because VQ indexing is a prerequisite needed to compute the statistics with which we generates tunings from. A model for how VQ codebook size impacts these statistics would allow for a completely hands-off, efficient, ANN solution. Additional work could refine the search cost model to more accurately reflect caches, account for network costs in distributed ANN solutions, and support alternative storage technologies such as flash. In general, we are also optimistic that the strategy of computing offline statistics from a query sample to model online ANN search behavior may generalize to non-quantization-based ANN algorithms as well.

ACKNOWLEDGMENTS

We would like to thank David Applegate, Sara Ahmadian, and Aaron Archer for generously providing their expertise in the field of optimization.

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

# A    APPENDIX

## A.1    LSH GUARANTEES IN PRACTICE

LSH based approaches to nearest neighbor search provide a number of rigorous guarantees regarding memory usage and query time complexity as a function of the approximation factor $c$, defined as follows: if the true nearest neighbor is a distance of $d$ from the query, the approximate algorithm is considered to have succeeded if it returns a point whose distance to the query is at most $cd$.

LSH algorithms are capable of prescribing hyperparameter settings (size and number of hash tables) to provide optimal worst-case performance for a given value of $c$. To see what these hyperparameter settings would imply for a given recall, we take a 90% recall@10 setting for `Glove1M` in Section 5.1 and compute the resulting $c$ for each query, giving us $10^4$ ratios. In Table 2 below, we plot various statistics of these ratios and their implications for the resulting ANN hyperparameters, using the results from Andoni & Razenshteyn (2015):

Table 2: Statistics on empirically measured $c$

| Statistic on $c$ | Value of $c$ | $\rho = 1/(2c^2 - 1)$ | Space Complexity | Estimated Space for `Glove1M` |
|---|---|---|---|---|
| Mean | 1.00262 | 0.98962 | $O(n^{1.99} + nd)$ | 1.2TB |
| Median | 1 | 1 | $O(n^2 + nd)$ | 1.4TB |
| 75th Percentile | 1 | 1 | $O(n^2 + nd)$ | 1.4TB |
| 90th Percentile | 1 | 1 | $O(n^2 + nd)$ | 1.4TB |
| 99th Percentile | 1.07024 | 0.77470 | $O(n^{1.77} + nd)$ | 60GB |

We see that even if we take the 99th percentile $c$ value, this leads to hash table hyperparameters that result in over 60GB of memory usage, under the conservative assumption that the constant factor for the $n^{1.77}$ term is one byte. This is 128 times the size of the original dataset and already somewhat impractical; for billions-scale datasets, this would be completely intractable.

While LSH may perform better in practice on this dataset than what the theory guarantees, and may not need such high memory consumption to reach this level of accuracy, such a result would imply that the hyperparameters for hashing-based ANN algorithms are difficult to tune and model, not unlike the hyperparameter-tuning challenges that other ANN algorithms face.

## A.2    FACTORIZED RECALL ASSUMPTION

Here we describe in detail the approximation $\mathbb{1}_{g \in \mathcal{S}_i(q,t)} \approx \mathbb{1}_{g \in \mathcal{S}_{i-1}(q,t)} \mathbb{1}_{g \in \mathcal{S}'_i(q,t_i)}$ used to simplify Equation 4. The event $g \in \mathcal{S}_i(q,t)$ is equivalent to requiring both $g \in \mathcal{S}_{i-1}(q,t)$ and finding fewer than $t_i$ points closer to $q$ than datapoint $g$ according to quantization $i$, which we can express as:

$$\mathbb{1}_{g \in \mathcal{S}_i(q,t)} = \mathbb{1}_{g \in \mathcal{S}_{i-1}(q,t)} \mathbb{1}\left\{ \sum_{j \in \mathcal{S}_{i-1}(q,t)} \mathbb{1}_{D(q,\tilde{\mathcal{X}}_j^{(i)}) < D(q,\tilde{\mathcal{X}}_g^{(i)})} < t_i \right\}. \tag{7}$$

Now define $\hat{\mathcal{S}}_{i-1}(q,t)$ to be $\{1, \ldots, n\} \setminus \mathcal{S}_{i-1}(q,t)$, and consider some element $j \in \hat{\mathcal{S}}_{i-1}(q,t)$. For the following section, we assume $g \in \mathcal{S}_{i-1}(q,t)$, which implies $D(q,\tilde{\mathcal{X}}_g^{(i-1)}) < D(q,\tilde{\mathcal{X}}_j^{(i-1)})$. We split our analysis into two cases:

1. $j \notin \mathcal{G}(q)$; this is by far the common case, where quantization $i - 1$ correctly removed a non-nearest-neighbor from the candidate set. This means quantization $i - 1$ has already ranked the relative distances of $j$ and $g$ correctly; given that quantization $i$ is higher bitrate than $i - 1$ and is therefore even less likely to mis-rank the relative distances, we know almost surely that $D(q,\tilde{\mathcal{X}}_g^{(i)}) < D(q,\tilde{\mathcal{X}}_j^{(i)})$.

2. $j \in \mathcal{G}(q)$; quantization $i - 1$ has dropped a nearest neighbor. If we assume $j$ and $g$ are picked uniformly without replacement from $\mathcal{G}(q)$, then with the original dataset $\mathcal{X}$,

$P[D(q, \mathcal{X}_g) < D(q, \mathcal{X}_j)] = 1/2$. However, at quantization $\tilde{\mathcal{X}}^{(i)}$, we approximate that $P[D(q, \tilde{\mathcal{X}}_g^{(i)}) < D(q, \tilde{\mathcal{X}}_j^{(i)})] = 1$ with the following justifications:

- Given that $D(q, \tilde{\mathcal{X}}_g^{(i-1)}) < D(q, \tilde{\mathcal{X}}_j^{(i-1)})$, $g$ is likely closer to $q$ than $j$ is, so the uniform sampling model used above for $g$ and $j$ leads to an underestimated probability.
- Even if $j$ is in fact the closer neighbor, the hierarchical nature of multi-layer quantization training implies that there is some correlation between distance mis-rankings among the different quantization layers. If $D(q, \tilde{\mathcal{X}}_g^{(i-1)}) < D(q, \tilde{\mathcal{X}}_j^{(i-1)})$ then there is a higher chance that $D(q, \tilde{\mathcal{X}}_g^{(i)}) < D(q, \tilde{\mathcal{X}}_j^{(i)})$.

Combining the results from our two cases, we see that if $g \in \mathcal{S}_{i-1}(q, t)$, then the number of elements in $\hat{\mathcal{S}}_{i-1}(q, t)$ that will rank closer to $q$ than $g$ according to $\tilde{\mathcal{X}}^{(i)}$ is approximately zero:

$$\mathbb{1}_{g \in \mathcal{S}_{i-1}(q,t)} \sum_{j \in \hat{\mathcal{S}}_{i-1}(q,t)} \mathbb{1}_{D(q,\tilde{\mathcal{X}}_j^{(i)}) < D(q,\tilde{\mathcal{X}}_g^{(i)})} \approx 0.$$

We can add this expression to Equation 7 and utilize the definition from Equation 3 to conclude

$$\mathbb{1}_{g \in \mathcal{S}_i(q,t)} \approx \mathbb{1}_{g \in \mathcal{S}_{i-1}(q,t)} \mathbb{1} \left\{ \sum_{j \in \mathcal{S}_{i-1}(q,t) \cup \hat{\mathcal{S}}_{i-1}(q,t)} \mathbb{1}_{D(q,\tilde{\mathcal{X}}_j^{(i)}) < D(q,\tilde{\mathcal{X}}_g^{(i)})} < t_i \right\}$$

$$= \mathbb{1}_{g \in \mathcal{S}_{i-1}(q,t)} \mathbb{1}_{g \in \mathcal{S}_i'(q,t_i)}.$$

## A.3 Extension of Cost Model to Batched Querying

The main speedup from query batching in quantization-based ANN algorithms arises from the fact that the same quantized data may be reused to score multiple queries. This conserves memory bandwidth, resulting in better performance, because memory bandwidth is scarce relative to arithmetic throughput in modern hardware. To account for this in our cost model, we first rewrite Equation 6 as

$$J(t) \triangleq \frac{1}{|\mathcal{X}|} \cdot \left( |\tilde{\mathcal{X}}^{(1)}| + \sum_{i=1}^{m-1} \frac{t_i}{n} \cdot |\tilde{\mathcal{X}}^{(i+1)}| \right)$$

$$= \frac{1}{|\mathcal{X}|} \sum_{i=1}^{m} \alpha_i |\tilde{\mathcal{X}}^{(i)}|$$

where $\alpha_i$ is the proportion of $\tilde{\mathcal{X}}^{(i)}$ that is searched; $\alpha_1 = 1$ and $\alpha_i = t_{i-1}/n$ for $i > 1$. The amount of memory bandwidth consumed in searching this quantization level for a batch of size $B$ is approximately $\min(1, \alpha_i B)|\tilde{\mathcal{X}}^{(i)}|$, while the number of arithmetic operations performed is $\alpha_i B|\tilde{\mathcal{X}}^{(i)}|$. Applying the roofline model to this algorithm, where we are bandwidth-bound in the small-batch regime and compute-bound in the large-batch regime, gives the following new cost model:

$$J(t, B) \triangleq \frac{1}{|\mathcal{X}|} \sum_{i=1}^{m} \max(\alpha_i B/\rho, \min(1, \alpha_i B))|\tilde{\mathcal{X}}^{(i)}|$$

where $\rho$ is the ratio of the hardware's arithmetic performance to memory bandwidth. In practice, query batching is of limited utility in large-scale ANN because $\alpha_i$ is so small that the algorithm will still be bandwidth-limited, resulting in no performance boost.

## A.4 Single-Layer Candidate Set Recall Curve Computation

Our goal is to efficiently compute $\mathcal{L}_i$ from Equation 5, reproduced below:

$$\mathcal{L}_i(\mathcal{Q}, t_i) = \mathbb{E}_{q \in \mathcal{Q}} \left[ -\log \frac{|\mathcal{S}'_i(q, t_i) \cap \mathcal{G}(q)|}{|\mathcal{G}(q)|} \right].$$

We want to compute this quantity for all $t_i \in \{1, \ldots, n\}$ and for all quantization levels. As a prerequisite, we first require the ground truth $\mathcal{G}(q)$ for all $q \in \mathcal{Q}$. Note this ground truth is required to compute the recall of any ANN algorithm and therefore should be computed regardless of whether our technique is used. We can store the ground truth results in a matrix $G \in \{1, \ldots, n\}^{n_q \times k}$, where $n$ is the number of elements in the dataset, $n_q$ is the number of elements in the $\mathcal{Q}$, and $k$ is the number of neighbors we want to retrieve per query.

Now for each ground truth element, we compute its distance from the query for every dataset quantization. This may be expressed as computing $U \in \mathbb{R}^{n_q \times m \times k}$ where $U_{a,b,c} = D(\mathcal{Q}_a, \tilde{\mathcal{X}}^{(b)}_{G_{a,c}})$. We then sort each $U_{a,b}$ so that $U_{a,b,c} < U_{a,b,c+1}$ for all $c \in \{1, \ldots, k-1\}$. From $U$ we then compute a matrix $V \in \mathbb{N}^{n_q \times m \times k}$ where $V_{a,b,c}$ stores the number of distances between query $\mathcal{Q}_a$ and $\tilde{\mathcal{X}}^{(b)}$ that are less than $M_{a,b,c}$:

$$V_{a,b,c} = \sum_{i=1}^{n} \mathbb{1}_{D(\mathcal{Q}_a, \tilde{\mathcal{X}}^{(b)}_i) < M_{a,b,c}}.$$

Note that because $U_{a,b}$ was sorted, $V_{a,b}$ will be sorted as well. The entry $V_{a,b,c}$ indicates the search depths at which recall for query $\mathcal{Q}_a$ with quantization $\tilde{\mathcal{X}}^{(b)}$ increase. For example, if $V_{a,b} = [1, 3, 8]$, we can conclude that recall for this query-quantization combination will be 0 when returning only the top 1 index, 1/3 when returning the top 2 or 3 indices, 2/3 when returning the top 3 through 7 indices, and 1 when returning the top 8 or more indices. Once we have $V$, it is a simple matter of taking logarithms and aggregating over queries to compute $\mathcal{L}_i$.

The computational bottleneck to this routine is computing $V$; we must compute the query-dataset distance for all queries over all dataset quantizations, which takes $O\left(n_q \sum_{i=1}^{m} |\tilde{\mathcal{X}}^{(i)}|\right)$ time, where $|\tilde{\mathcal{X}}^{(i)}|$ denotes the memory footprint of quantization $i$. For each distance we must then decide how many of $M_{a,b}$ it is less than, which takes $O(\log k)$ time per distance using binary search. We can estimate the number of distances as $O(mn)$, although this is an overestimate because VQ-quantized layers will result in fewer than $n$ distances being materialized. Overall, this gives our routine a runtime of

$$O\left(n_q \sum_{i=1}^{m} |\tilde{\mathcal{X}}^{(i)}| + n_q m n \log k\right).$$

To analyze this expression further, let $|\tilde{\mathcal{X}}^{(i)}| = n \cdot d \cdot r_i$, where $r_i$ is the compression ratio for quantization $i$. Our runtime can then be expressed as $O(n_q n(d \sum_{i=1}^{m} r_i + m \log k))$; the first term tends to dominate due to the presence of $d$.

This routine is relatively cheap; to compare, the computation of ground truth nearest neighbors, which is generally done as part of any ANN indexing routine, takes $O(n_q n d)$ time. Assuming as above that the first term of our runtime dominates, our routine is a multiplicative factor of $\sum_{i=1}^{m} r_i$ as expensive, which in practice with typical quantization hierarchies is below 2.

## A.5 FAST MINIMIZATION ALGORITHM FOR DYNAMIC LAGRANGE MULTIPLIERS

Our goal is to quickly perform the optimization

$$\underset{t \in [0,n]^m}{\arg\min} \quad -\lambda J(t) + \sum_{i=1}^{m} \mathcal{L}_i(t_i)$$
$$\text{s.t.} \quad t_1 \geq \ldots \geq t_m.$$

quickly for dynamic values of $\lambda$, with all other inputs constant. We first describe the zero-preprocessing, $O(nm)$ time dynamic programming solution to this problem, which we then optimize to arrive at our final $O(m \log n)$ approach.

### A.5.1 BASIC DYNAMIC PROGRAMMING APPROACH

For a fixed $\lambda$, our optimization is equivalent to selecting non-increasing indices $t_1, \ldots, t_m$ such that $\sum_{i=1}^{m} M_{i,t_i}$ is minimized, where the matrix $M \in \mathbb{R}^{m \times n}$ is defined as $M_{i,j} = \mathcal{L}_i(j) - \lambda J_i(j)$. Now we define our dynamic programming subproblem as selecting the first $a$ indices $t_1, \ldots, t_a$ such that all indices are at least equal to $b$. We can store these subproblem answers in another matrix $M'$:

$$M'_{a,b} \triangleq \min_{t \in [b,n]^a} \quad \sum_{i=1}^{a} M_{i,t_i}$$
$$\text{s.t.} \quad t_1 \geq \ldots \geq t_a.$$

Using the state transition $M'_{a,b} = \min(M'_{a,b+1}, M_{a,b} + M'_{a-1,b})$ allows us to compute all of $M'$ and receive the resulting minimum, located at $M'_{m,n}$, in $O(nm)$ time. The approach may be augmented with cost and history matrices so that $J(t), \sum \mathcal{L}_i$, and the selected indices $t_i$ may be recovered as well.

### A.5.2 FASTER ALGORITHM LEVERAGING CONVEXITY

Recall that all $\mathcal{L}_i$ are convex, and $J$ is linear, so every row of $M$ is convex. Inductively, we can see that this leads to the rows of $M'$ being convex as well, which we take advantage of to accelerate our algorithm. First, define $j^*(i)$ as the rightmost index in row $i$ achieving the row-wise minimum in $M'$; $j^*(i) = \max\{j : M'_{i,j} = \min_k M'_{i,k}\}$. Due to the convexity of $M'$, we know that $M'_{i,j}$ is strictly increasing with respect to $j$ for all $j > j^*(i)$. Combined with this equivalent expression for computing $M'$,

$$M'_{a,b} = \min_{j \in \{b, \ldots, n\}} M_{a,j} + M'_{a-1,j} \tag{8}$$

it's clear that $M'_i$ is equivalent to $M_i + M'_{i-1}$ in the column range $[j^*(i), n]$. Meanwhile, $M'_{i,j} = M'_{i,j^*(i)}$ for $j < j^*(i)$.

Now let's inductively assume that $M'_{i-1}$ is composed of a number of piecewise components, where the $k$th component equals the sum of the most recent $r_k$ rows of $M$ plus a constant $c_k$ and is responsible for the columns $j \in [l_k, r_k]$. From Equation 8 we can see that upon the transition from $M_{i-1}$ to $M'_i$, all pieces to the right of $j^*(i)$ will be incremented by $M_i$. All pieces left of $j^*(i)$ will be replaced with a constant offset equalling $M_{i,j^*(i)}$, maintaining our inductive hypothesis. The base case for this hypothesis is also easily verified, as we see $M'_1$ equals $M_1$ in the range $j \in [j^*(1), n]$, and left of that range it equals the constant $M'_{1,j^*(1)}$.

This realization allows for faster approaches to our minimization problem by allowing $M'$ to be implicitly represented rather than explicitly computed. We maintain the set of piecewise components as we traverse from row 1 to $m$. At each row, we have to find $j^*(i)$ and update our set of components accordingly. The search for $j^*(i)$ may be done efficiently by taking advantage of the convexity of $M'$ via binary search. We first binary search over our components to find the one containing the global minimum, and then binary searching among the indices belonging to that component. Computing the value within a component may be done in $O(1)$ time once a prefix sum array over $\mathcal{L}$ and $J$ are created; these data structures allow contiguous sums over $M$ with dynamic $\lambda$. The component list update then takes $O(1)$ amortized cost per row, because only one component is added per row.

There are at most $\min(m + 1, n)$ components and any component may have a range of at most $n$ indices, so the resulting minimization takes $O(m \log n)$ time. Calculating the prefix sums over $\mathcal{L}$ and $J$ result in $O(mn)$ preprocessing time for this algorithm.

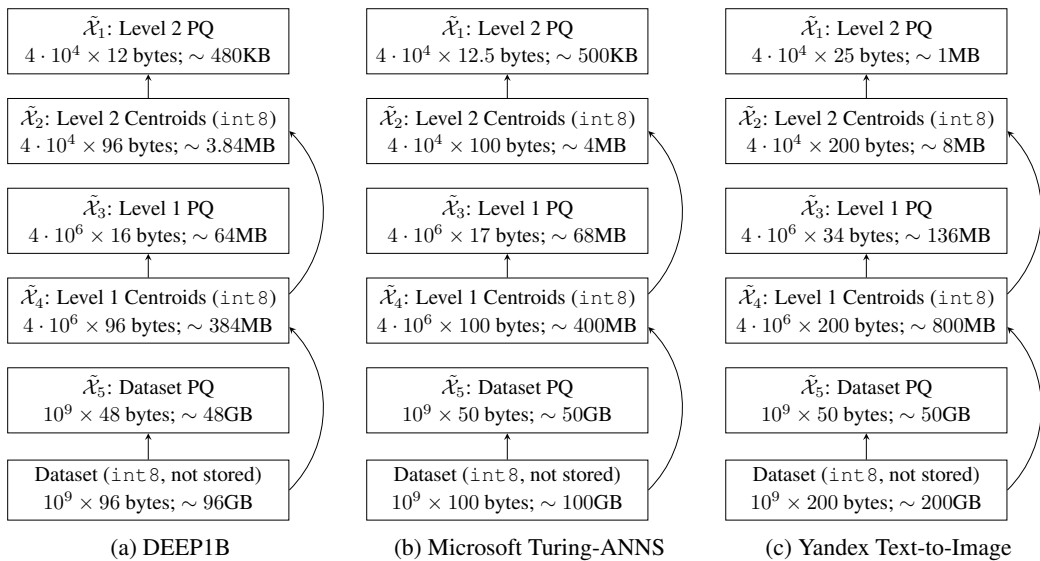

Figure 6: Multi-level quantization hierarchies used for the three billions-scale datasets of Section 5.2.

### A.6 BILLION-SCALE SEARCH EXPERIMENTAL SETUP DETAILS

Track 1 of the `https://big-ann-benchmarks.com` competition stipulates that:

- Query-time benchmarking is done on Azure VMs with 32 vCPUs and 64GB RAM.

- Indexing takes place on Azure VMs with 64vCPUs, 128GB RAM, and 4TB SSD, and must finish within 4 days.

Our query-time serving was performed on 16 physical cores from an Intel Cascade Lake-generation CPU, and peak memory usage was measured to be under 58GB. Azure counts one physical core as equal to 2 vCPUs, so our setup matches the competition requirements.

For indexing time, our index was constructed via a distributed computing pipeline. For DEEP1B, the pipeline ran in approximately 1.5 hours and in total consumed approximately 538.0 vCPU-hours of compute power. Approximately 16.3 vCPU-hours, or 2.7% of the overall indexing job's resource consumption, was spent on computing $\mathcal{L}$ for use in our convex optimization routine. Other datasets were comparable in runtime. They were all significantly under the 6144 vCPU-hours available under the competition specifications, and furthermore the VQ and PQ training involved in the pipeline have low peak memory and disk requirements, so our indexing procedure comfortably qualifies under the competition rules.

### A.6.1 HIERARCHICAL QUANTIZATION INDEX DETAILS

The three diagrams below describe in detail the quantization hierarchy used for the three billions-scale datasets in Section 5.2. We would like to emphasize that all three datasets used the same VQ and PQ settings–the datasets are first quantized to $4 \times 10^6$ centroids, and then again to $4 \times 10^4$ centroids. The PQ was performed with 16 centers (4 bits) per subspace, 4 dimensions per subspace at the $\mathcal{X}_1$ level, and 3 dimensions per subspace (rounding up) at the $\mathcal{X}_3$ level. Only the Yandex Text-to-Image dataset differs at the $\mathcal{X}_5$ level with its PQ setting (using 2 dimensions per subspace instead of 1), but this was only to fit the higher-dimensional dataset into the same RAM footprint.

Even though our technique cannot set these VQ and PQ parameters, the fact that we can achieve excellent performance on all three datasets using the same VQ and PQ parameters, despite these datasets giving drastically different speed/recall Pareto frontiers, suggests the VQ and PQ parameters are not as difficult a part of the tuning problem and have more predictable good settings than the hyperparameters our technique deals with.

## A.7 IMPACT OF QUERY SAMPLE SIZE ON HYPERPARAMETER TUNING QUALITY

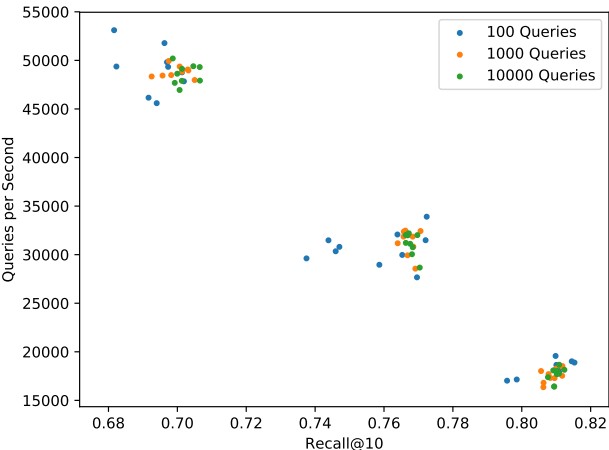

Figure 7: Achieved search throughput and recall for tunings generated from different query sample sizes on the Microsoft Turing-ANNS billion-scale dataset.

A larger query sample should lead to our optimization finding better hyperparameter tunings, but would also lead to greater cost in computing $\mathcal{L}$, whose runtime is linear in the query sample size. In this section, we explicitly measure the effect of increasing query sample size by taking the $10^5$ queries from the MS-Turing dataset of `big-ann-benchmarks`, holding out $10^4$ queries as a test set, and sampling subsets of 100, 1000, and 10000 queries from the remaining $9 \times 10^4$ queries. For each subset size, nine samples were taken, all disjoint from other subsets of the same size.

For each query subset, we run our constrained optimization technique to generate low, medium, and high search cost ANN algorithm hyperparameter tunings, and measure the resulting recall and search throughput on the holdout query set. The search cost targets for the low, medium, and high settings were $J(t) = 2 \times 10^{-6}, 5 \times 10^{-6}$, and $1.2 \times 10^{-5}$, respectively. The resulting recalls and throughputs are plotted in Figure 7.

We can see that increasing the query sample size from 100 to 1000 leads to considerably less variance, with the resulting hyperparameters more consistently approaching the global cost-recall Pareto frontier. However, increasing the sample size further from 1000 to 10000 has very marginal impact. The query sample size was kept at 1000 for the experiments in Section 5.2, and is reflected in the vCPU-hour measurement in Appendix A.6.

## A.8 PERFORMANCE OF FEW-LEVEL QUANTIZATION HIERARCHIES ON DEEP1B

In Section 5.2 we use a five-level quantization hierarchy to search the DEEP1B, Microsoft Turing-ANNS, and Yandex Text-to-Image datasets from `big-ann-benchmarks`. Our hyperparameter tuning technique was necessary because the four-dimensional hyperparameter search space was impractically large for grid search. If we had built an ANN search index with fewer quantization levels, a simpler grid search approach could be used to find effective hyperparameters; in this section, we confirm that such shallow search indices perform poorly on DEEP1B, thereby ruling out grid search as a viable option.

To test this, we take the original quantization hierarchy, described in Figure 6, and modify it in two ways. In one, we keep only the top-level $4 \times 10^4$ VQ centroids and remove the intermediate layer of $4 \times 10^6$ centroids, to create the `Shallow-Small` quantization index; in the other, we remove the top-level centroids and only keep the $4 \times 10^6$ centroids from the middle layer of the hierarchy. These two modified quantization indices are illustrated in Figure 8.

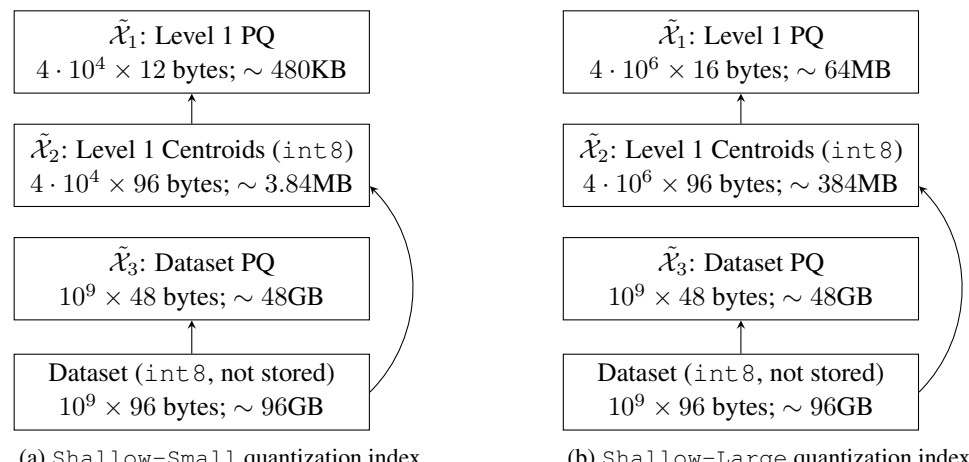

(a) Shallow-Small quantization index

(b) Shallow-Large quantization index

Figure 8: The two quantization hierarchies we compare in this experiment against the original one used in Section 5.2 and described in Figure 6.

The Shallow-Small and Shallow-Large indices both have three quantization levels and therefore a two-dimensional hyperparameter search space. We use grid search to explore this space, and plot in Figure 9 the resulting speed-recall Pareto frontiers against the original frontier from Section 5.2.

We see that both Shallow-Small and Shallow-Large perform extremely poorly relative to the original five-layer index. Qualitatively, Shallow-Small has too many datapoints assigned to each of its $4 \times 10^4$ centroids, which implies the search cost of further searching any centroid is high; in addition, any given datapoint is unlikely to be quantized well by its centroid (due to the relatively low number of centroids), so the quality of results also tends to be low. Meanwhile, Shallow-Large has so many centroids that it always spends a large fixed cost on level 1 PQ distance computation.

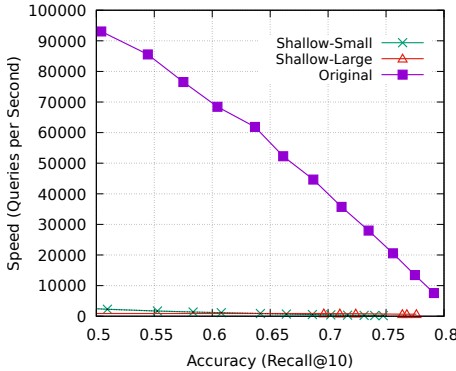

Figure 9: Both three-level quantization indices perform very poorly relative to the original five-layer quantization index on DEEP1B.

