# OpenReview forum: "Automating Nearest Neighbor Search Configuration with Constrained Optimization"
_ICLR.cc/2023/Conference — ICLR 2023 poster_

### Official Review · Reviewer_DrxR · 2022-10-18

**Confidence:** 4
**Correctness:** 4
**Technical Novelty And Significance:** 4
**Empirical Novelty And Significance:** 4
**Recommendation:** 8

**Clarity, Quality, Novelty And Reproducibility:**

The paper is very clearly written despite its technical content.  It address a problem that I suspect is of considerable importance, namely approximate nearest neighbour detection.  This is not my field, but this seems to address an important problem of existing techniques is that it is hard to find appropriate hyper-parameters to make these methods work for different types of dataset.  The approach used seems novel.  The results seem reproducible.

**Strength And Weaknesses:**

This is a well written paper.  The idea is elegant and sophisticated.  The empirical evaluation looks very convincing.

Although this is not my research field I see no obvious weaknesses.  Two very minor points that can be easily corrected.  The first is the Pareto should be capitalised as it refers to a proper name (Vilfredo Pareto).  Secondly, I might be wrong but I could not see a reference to Figure 2 in the text.  I think this should be referenced in Section 5.2.

**Summary Of The Paper:**

The paper covers a new method for approximate nearest neighbour search aimed at very large datasets where hyper-parameter optimisation can be achieved efficiently.  The hyper-parameter optimisation uses some clever approximations that then allow very efficient optimisation.  The performance is measured empirically and shown to give better results than current methods.

**Summary Of The Review:**

This is a well written paper with a nice idea that looks well motivated and technically sound.  The empirical results back up the claims (there is one approximation about independence that needs empirical justification).  In my view this is a good paper that deserves publication.

---

> ### Author Response · Authors · 2022-11-10
> **Response to reviewer DrxR**
>
> Thank you for the review! In response to your points, we’ve capitalized all references to the word “Pareto” and added an explicit reference to Figure 2 in Section 5.2.

---

### Official Review · Reviewer_cFiP · 2022-10-24

**Confidence:** 3
**Correctness:** 3
**Technical Novelty And Significance:** 4
**Empirical Novelty And Significance:** 3
**Recommendation:** 8

**Clarity, Quality, Novelty And Reproducibility:**

Overall, based on the strength and weaknesses of this paper, the clarity and the quality seem good to me.

Considering the proposed approach is computationally efficient and has a theoretical guarantee, it seems to me that this paper is novel.



**Details Of Ethics Concerns:**

No.

**Strength And Weaknesses:**

The paper has the following strengths:

S1. The motivation is clear. As the dataset size increases significantly, tuning the parameters (manually or using traditional methods like grid search and black box optimizer) becomes computationally prohibitive. It is in significant demand to have a theoretically grounded yet computationally lightweight approach that can produce excellent performance.

S2. The proposed method is novel and easy to use. The input is simply the search cost or recall, which is friendly for fresh users unfamiliar with the ANN methods' details. The authors also provide sufficient insights and theoretical analysis when they introduce their idea.

S3. This paper is well written with clear clarity and thus is easy to read.

The paper has the following weaknesses:

W1. In the last paragraph of the introduction section, they claim their approach is very general and can be extended to distance measures, quantization algorithms, and search paradigms. It would be nice if they could simply illustrate how to extend such a claim, e.g. for different search paradigms.

W2. In Section 5.2, they claim they compare to heuristic, hand-tuned, and black-box optimizer settings. However, I cannot find such settings in Figure 2, and Figure 3 only depicts the comparison with a black-box optimizer Vizier. Moreover, the baselines that are shown in Figure 2 lack explanations.

W3. In Section 5.3 and Figure 4, could you explain what $R^2$ is?

**Summary Of The Paper:**

This paper looks at the problem of hyperparameter tuning for the Approximate Nearest Neighbor (ANN) search. The authors propose a constrained optimization-based approach for tuning the quantization-based ANN methods. They only require a search cost or recall as input, then the Lagrange multipliers-based approach can provide a configuration such that the performance reaches the speed-recall pareto frontier. Extensive results confirm the superior performance of their proposed method.


**Summary Of The Review:**

In summary, I think the motivation is clear, the idea is sound, and the paper is written well enough. Thus, I tend to initially give acceptance to this paper.

---

> ### Author Response · Authors · 2022-11-10
> **Response to reviewer cFiP**
>
> Thank you for the review! Below is our response to your points W1-W3:
>
> 1. For example, to extend to a new distance measure, notice that the computation of $\mathcal{L}$ is agnostic to the actual distance measure. We narrow down our candidate set by choosing the vectors $x_i$ that minimize $D(q, x_i)$ for a distance $D$, and the recall from these candidate sets is used to compute $\mathcal{L}$, so even though our experiments were conducted on Euclidean distance datasets, any other distance measure could be substituted in.
> 1. The comparisons in Figure 2 were taken from [here](https://github.com/harsha-simhadri/big-ann-benchmarks/blob/main/t1_t2/results/T1/neurips21/t1.csv) which should be linked from the original big-ann-benchmarks website. One of the columns of that CSV describes the configurations, which should be interpretable when combined with the Python wrapper code for each submission ([example](https://github.com/harsha-simhadri/big-ann-benchmarks/pull/69/files)).
> 1. We were referring to the square of the sample Pearson correlation coefficient, which is 1 for perfectly linear relationships. We’ve updated the text in Section 5.3 and Figure 4 to emphasize this.

---

### Official Review · Reviewer_F6KH · 2022-10-25

**Confidence:** 4
**Correctness:** 4
**Technical Novelty And Significance:** 3
**Empirical Novelty And Significance:** 3
**Recommendation:** 6

**Clarity, Quality, Novelty And Reproducibility:**

**Clarity**
- What are the family of ANN algorithms that you are able to tune with your approach? linear scan? IVF? hashing? or just linear scans?
- If the answer to above question is linear scan, I suppose the algorithm does a full scan over the entire billion scale dataset in 16bytes precision? Is it possible to do so in an interactive mode (one query at a time), or must the entire dataset be batch processed to get reasonable query throughput? Scanning 16GB (1B points X 16byte PQ) for each query seems expensive!
- In Alg 1 and Section 3.4 and 4, you list S1/X1 as the largest set with lowest bitrate. However, In fig 6, you list X5  as the largest set with highest bitrate. As in X5 has all the elements and has the highest number of bytes in its representation. Is there a mistake in Fig 6? If not, the exposition in 3.4 and 5 seems to contradict what is presented in Fig 6.

**Quality**
- In section 5.4, why not use Yandex T2I dataset which has out-of-distribution queries?
- Can you comment on how well your techniques apply to graph algorithms and other algorithm on Track 2/3 of the NeurIPS competition.
- Can you list the total number of hyperparameter configurations that can be chosen from at billion scale to give the reader a sense of the size of this space.
- If Google Vizier is allowed to run for more than 6 hours, does it find the pareto-optimal configurations found by your algorithm? Could you also comment a bit more about where Vizier would fail and where it might succeed?
- Can one sample the dataset (1B->1M), run exhaustive tuning to find 4 levels of quantization and then train the 5th level while going back up to 1B? How much worse would this be compared to one-shot tuning using your proxy loss.

**Reproducibility**
-- Are the results on the training query set or the validation query set released in the competition?



**Strength And Weaknesses:**

Strengths:
1. Good empirical results -- improves significantly on baselines, outperforms blackbox tuning (e.g. Vizier) and on small scale datasets closely matches exhaustive tuning
2. Results demonstrated on multiple large scale datasets.


Weakness:
1. Not much discussion on applicability to tuning algorithms other than the linear scan in the paper.
2. Its not clear that the techniques can be applied for interactive query processing.

**Summary Of The Paper:**

ANNS algorithms, especially those involving quantization, have many hyperparameters to tune. Careful tuning can yield significantly better accuracy and query throughput. However, tuning can be expensive, especially for large indices, if done naively (build the index and search). Therefore, the authors design a proxy loss metric to guide the hyperparameter search using a simple linear-scan based algo. This loss is an approximation of the metrics that matter in practice. Using this they show that on billion scale datasets, the tuning can outperform baselines by a margin and improve upon blackbox optimizers. Further, the authors entry won Track 1 of a recent NeurIPS challenge on large scale ANNS.

**Summary Of The Review:**

The results presented in the paper are a great improvement over baselines, however the paper could be stronger if it generalizes to interactive query processing or to other algorithms rather than the linear scan. It could be that the techniques are applicable, but couldn't ascertain this in my reading of this paper. I would rate the paper higher if this was the case.

---

> ### Author Response · Authors · 2022-11-10
> **Response to reviewer F6KH**
>
> Thank you for the review! To summarize our response to the two weaknesses mentioned, our technique works for quantization-based ANN algorithms, a fairly broad class, and our technique is in fact meant for ANN search systems that perform online, interactive query processing. Full response to clarity/quality/reproducibility points below:
>
> Clarity
> * Section 3.4 describes the ANN algorithms supported by the current implementation of our technique. They can be summarized as algorithms using any combination of IVF and (potentially quantized) linear scan. We primarily refer to IVF as VQ, and Section 3.2 clarifies the relationship between these terms. We now explicitly reference the term “IVF” in the updated version of the paper (original version only referred to VQ as an “inverted index.”)
> * The algorithm does **not** do a full scan in 16 byte precision, and the queries do not have to be batch-processed. For an illustration of how the algorithm operates (for instance on DEEP1B), please refer to Algorithm 1 and Figure 6a. The algorithm will start with a linear scan over $X_1$, which is only around 480KB (this is the size of approximately 1000, not 1 billion, unquantized datapoints). From there, the higher-bitrate quantizations are only partially searched due to the reduction in $S$.
> * We believe our notation is correct; $X_1$ is the lowest bitrate, and $X_m$ is the highest bitrate; it therefore makes sense in Figure 6 that $X_5$ has the highest number of bytes in its representation. Meanwhile, $\mathcal{S}$ becomes progressively smaller as the candidate set is narrowed down.
>
> Quality
> * Our intent was to test performance on queries in-distribution, but out of the training sample, and therefore not identical, to the queries used to compute $\mathcal{L}$. We feel this is a more realistic setup; without an in-distribution assumption, any model used to generate the vectors likely also cannot provide any guarantees about correspondence between vector distance and underlying semantic similarity, so ANN results may become meaningless. It is very difficult to provide out-of-distribution generalization guarantees in machine learning in general.
> * We are not experts in the implementations submitted to T2 and T3, but below we give a brief overview based on our understanding. Some submissions, such as the `faiss_t3` baseline and `cuanns_ivfpq` submission to T3, can directly be characterized as hierarchical quantization indices (Section 3.4) and therefore should work fairly straightforwardly (perhaps with some adjustments to the cost model $J$ for memory accesses in a different architecture, ex. Appendix A.3). Graph-based algorithms require more adjustments; a method of factorizing recall into a search list size component and beam search width component would be necessary, and $J$ would have to model the beam width’s impact on performance (random read queue depth has a beneficial but nonlinear impact on throughput).
> * We mention in Section 5.2 that “Even with very optimistic assumptions, this gives hundreds of thousands of tunings to grid search over, which is computationally intractable.” Here is some back-of-the-envelope math for this number: for the billions-scale datasets, we have a five-dimensional hyperparameter vector $t$, where we know $t_5=10$ when we want the 10 nearest neighbors; this gives us a four-dimensional hyperparameter space. We require $10^9\ge t_1\ge t_2\ge t_3\ge t_4\ge t_5=10$. To reduce the search space, we could use a geometrically spaced grid; assuming a spacing of 10%, we’d have a search space of $[\log_{1.10}(10^9/10)]^4 / 4!$, or roughly 58 million. Even if there were some heuristics to rule out 99% of this search space, we’d have 580K possibilities left.
> * We found the computational costs of running the Vizier approach further to be fairly high, so we didn’t run beyond 6 hours. Vizier didn’t seem to learn a good model for predicting ANN recall from hyperparameter settings, and was also likely hindered by measurement noise in QPS measurements (due to machine performance fluctuations, which are gradually smoothed out with longer measurements, with the tradeoff of making trials slower).
> * This would likely provide similar quality tunings at the cost of significantly more computational cost. We’d first have to do the grid search to tune the upper quantization levels; then, for any given target throughput, we’d have to do another iso-throughput linear sweep through (low search effort in upper quantization levels, high search effort in last quantization level) to (high search effort in upper quantization levels, low search effort in last quantization level) to find which provides maximum recall. This may be computationally feasible but is noticeably more expensive, and doesn’t scale as datasets grow larger and more quantization levels are used.
>
> Reproducibility: The queries under the “Query data” column of the table in the “Benchmark Datasets” section of https://big-ann-benchmarks.com/ was used.

---

### Official Review · Reviewer_vMnt · 2022-10-25

**Confidence:** 4
**Correctness:** 4
**Technical Novelty And Significance:** 3
**Empirical Novelty And Significance:** 3
**Recommendation:** 5

**Clarity, Quality, Novelty And Reproducibility:**

The paper is written well and clearly. The method is novel. No code is provided for reproducibility

**Strength And Weaknesses:**

Strengths:

1) The paper is novel and shows a performance improvement over baselines with negligible optimization cost.
2) The optimization cost is carefully designed. The method is based on maximizing the geometric mean recall for maximizing the ANN recall. It maximizes the arithmetic mean lower bound. To minimize the ANN search time, it minimizes memory accesses during the query phase. The overall constrained optimization maximizes the recall with the given search time constraints.
3) The authors have shown the benchmark against the grid search and one black box search method- Vizier.

Questions and weaknesses:
1) Overhead of the parameter tuning: How does the parameter tuning time compare with the Black- box optimizers such as Vizier?
2) What is the rationale behind choosing the recall rate as the objective?:  The objective function is the recall rate, and the constraint is the search cost. Does the opposite, where the objective function is the search cost, and the constraint is the recall rate, results in a similar performance?
3) Please correct me if I am wrong here: What query set is used for computing the objective function- $\mathcal{L}_i (t_i)$? As mentioned in 5.4, it looks like the query set is being used for optimization. However, this will not suit well for the real-time near-neighbor search, where the queries are unseen and seen one-by-one or batch-by-batch as a stream. The optimization cost should be negligible to achieve a real-time performance here.
4) With an assumption of an in-distribution query set (same distribution of index and query points), this can even use a fraction of index data to optimize the parameters.
5) Are there any other black box optimizers available for the ANN parameter tuning?
6) Section 5.4: The random split of the query set $Q_1$, and $Q_2$ is not the best way to see the out-of-sample performance. A better split where $Q_1$ and $Q_2$ are itself out of distribution to each other may be a better way to test the out-of-sample performance. Please clarify if I am missing something here.

**Summary Of The Paper:**

The paper uses the recursive vector quantization ANN method, which can also be seen as a hierarchical VQ approach. The proposed method provides an optimization-based approach to fine-tune this ANN method. The analysis was made on million-scale and billion-scale ANN datasets.

**Summary Of The Review:**

The constraint optimization formulation is based on the search time and recall is intuitive, however, the applicability of the optimizer is questionable for the unseen query data.

---

> ### Author Response · Authors · 2022-11-10
> **Response to reviewer vMnt**
>
> Thank you for the review! Below is our response to your points:
>
> 1. We ran Vizier for 6 hours on a machine with 16 physical cores, so 192 vCPU-hours. Our technique took 16.3 vCPU-hours to compute $\mathcal{L}$ and under one second to generate tunings by applying the algorithm in Appendix A.5.2 to $\mathcal{L}$. These numbers are mentioned in Section 5.2, Section 4.4, and Appendix A.6, respectively.
> 1. The technique performs identically when aiming to minimize cost with a constraint on minimum recall. The algorithm changes very little; we would still binary search on $\lambda$ as described in Section 4.4, but we would compare achieved recall to the recall minimum instead of comparing achieved cost to the cost maximum when updating the binary search range.
> 1. The query set should be a representative sample, taken offline, of the query stream. The entire procedure of generating $\mathcal{L}$ and an ANN hyperparameter tuning should also be done offline, in conjunction with the training of the index. There are therefore no real-time performance issues, because when a query comes in, we simply leverage the ANN algorithm to perform search as usual; the tuning was configured beforehand.
> 1. We found in Appendix A.6 that our procedure only takes 2.7% of the overall ANN indexing resource usage, which was low in both absolute and relative terms, so we did not optimize further, but we agree some sampling of the database could provide speedups if further indexing optimizations must be done.
> 1. We believe many black-box optimizers could’ve been used in place of Vizier for our experiment in Section 5.2.1, but we were most familiar with Vizier, and as far as we understood, Vizier generally provides performance competitive with other black-box optimizers, so that’s what we chose.
> 1. Please let us know if you believe our terminology in Section 5.4 can be better phrased, but our intent was to test performance on queries in-distribution, but out of the training sample and therefore not identical, to the queries our technique used to compute $\mathcal{L}$. We feel this is a more realistic setup; without an in-distribution assumption, any model used to generate the vectors likely also cannot provide any guarantees about correspondence between vector distance and underlying semantic similarity, so ANN results may become meaningless. It is very difficult to provide out-of-distribution generalization guarantees in machine learning in general.

---

### Author Response · Authors · 2022-11-10
**Rebuttal**

Hi all, thank you for taking the time to review our work. We have just uploaded a rebuttal revision of our paper that addresses a number of points made in the reviews, and we furthermore have uploaded an implementation of the algorithms described in Appendix A.5.1 and A.5.2 as supplementary materials.

Our responses to your individual reviews detail the changes to the paper we've made, and should address your other questions/concerns--thank you again for your time.

---

### Decision · Program_Chairs · 2023-01-20

**Decision:**

Accept: poster

**Justification For Why Not Higher Score:**

The paper shows the novelty on ANN search problems. But, reviewers find many issues that need to be fixed.

**Justification For Why Not Lower Score:**

N/A

**Metareview: Summary, Strengths And Weaknesses:**

This work proposes a proxy loss metric to guide the hyperparameter for ANN search problems. The idea is nice, and the experiments have demonstrated the good performance of the proposed method. All the reviewers find the novel contribution of this paper, and are all positive to this paper. Thus, I recommend accept.

**Note From Pc:**

if the above contains the word "oral" or "spotlight" please see: "oral" presentation means -> notable-top-5% and "spotlight" means -> notable-top-25%. As stated in our emails, we are disassociating presentation type from AC recommendations